# The curious case of the Dana platypus and what it can teach us about how lead shotgun pellets behave in fluid preserved museum specimens and may limit their scientific value

Henrik Lauridsen[1]*, Daniel Klingberg Johansson[2], Christina Carøe Ejlskov Pedersen[3], Kasper Hansen[3], Michiel Krols[4], Kristian Murphy Gregersen[5], Julie Nogel Jæger[5], Catherine Jane Alexandra Williams[6], Ditte-Mari Sandgreen[7], Aage Kristian Olsen Alstrup[1,8], Mads Frost Bertelsen[9,10], Peter Rask Møller[2]

1 Department of Clinical Medicine, Aarhus University, Aarhus, Denmark, 2 Natural History Museum of Denmark, University of Copenhagen, Copenhagen, Denmark, 3 Department of Forensic Medicine, Aarhus University, Aarhus, Denmark, 4 TESCAN XRE, Ghent, Belgium, 5 Institute of Conservation, Royal Danish Academy, Copenhagen, Denmark, 6 Department of Animal and Veterinary Sciences, Aarhus University, Aarhus, Denmark, 7 Givskud Zoo-Zootopia, Give, Denmark, 8 Department of Nuclear Medicine & PET, Aarhus University Hospital, Aarhus, Denmark, 9 Center for Zoo and Wild Animal Health, Copenhagen Zoo, Frederiksberg, Denmark, 10 Department of Veterinary Clinical Sciences, University of Copenhagen, Copenhagen, Denmark

* henrik@clin.au.dk

**Data Availability Statement:** Raw CT, micro-CT and MRI data of the Dana platypus NHMD-M01-28

## Abstract

Fluid preserved animal specimens in the collections of natural history museums constitute an invaluable archive of past and present animal diversity. Well-preserved specimens have a shelf-life spanning centuries and are widely used for e.g. anatomical, taxonomical and genetic studies. The way specimens were collected depended on the type of animal and the historical setting. As many small mammals and birds were historically collected by shooting, large quantities of heavy metal residues, primarily lead, may have been introduced into the sample in the form of lead shot pellets. Over time, these pellets may react with tissue fluids and/or the fixation and preservation agents and corrode into lead salts. As these chemicals are toxic, they could constitute a health issue to collection staff. Additionally, heavy element chemicals interfere with several imaging technologies increasingly used for non-invasive studies, and may confound anatomical and pathological investigations on affected specimens. Here we present a case-study based on platypus (*Ornithorhynchus anatinus*) and other small mammals containing lead pellets from the collection of The Natural History Museum of Denmark.

## Introduction

The first recorded mention of the use of high concentrations of ethanol to preserve whole animal specimens was on June 4, 1662, when the physician William Croone appeared before the

is available for download on MorphoSource (https://www.morphosource.org/) project# 000620016 using the link: https://www.morphosource.org/projects/000620016?locale=en. Raw digital x-ray images of remaining specimens can be downloaded from Figshare (https://figshare.com/) using the link: https://doi.org/10.6084/m9.figshare.25569099.v1.

**Funding:** HL is supported by the Carlsberg Foundation (grant# CF21-0605) and PRM is supported by the Carlsberg Foundation (grant# CF21-0435). The funders had no role in study design, data collection and analysis, decision to publish, or preparation of the manuscript.

**Competing interests:** I have read the journal's policy and the author Michiel Krols of this manuscript has the following competing interests: Employed at TESCAN XRE, a manufacture of one of the micro-CT systems used in the study (UniTOM XL Spectral). The remaining authors have declared that no competing interests exist. This does not alter our adherence to PLOS ONE policies on sharing data and materials

Royal Society of London and "produced two embryos of puppy-dogs, which he had kept eight days, and were put in spirit in a glass-vial sealed hermetically" [1–3]. Since then, ethanol preservation has remained a common practice for the long-term storage of animal specimens in the collections of natural history museums around the globe. When the fixative powers of formaldehyde were described in the late 19th century by another physician, Ferdinand Blum [4, 5], preservation practices were expanded to often include an initial fixation step by injecting or immersing animal specimens in 4% v/v formaldehyde followed by transferring to gradually higher concentrations of alcohols e.g. ethanol (EtOH) until usually reaching 70% v/v [3]. In this state, preserved animal specimens can be stored for centuries and constitute important and often irreplaceable archives of past and present biodiversity [6].

A useful aspect of museum collections of fluid preserved animals is the possibility to conduct both patho-anatomical and comparative anatomy studies across a wide range of species. With the advent of modern medical imaging techniques such as x-ray computed tomography (CT) and magnetic resonance imaging (MRI) and their introduction in the zoological field, non-invasive/destructive studies are now possible on both soft, hard and fossilized tissues [7–14]. This adds considerable scientific value to collections of fluid preserved animals and drives present digitization efforts of such collections e.g. the Danish Distributed System of Scientific Collections (DaSSCo), the European Distributed System of Scientific Collections (DiSSCo), and the North American Open Exploration of Vertebrate Diversity in 3D (oVert) [6, 15].

Collection methods of museum specimens vary by the type of animal and the historical setting. However, to gain maximal scientific value of preserved specimens, the collection method can be of great importance. In the following, we describe a case study on a specimen of platypus *Ornithorhynchus anatinus* (Shaw, 1799), in which the discovery with x-rays of widely distributed dense nodules hugely impacted pathological and anatomical interpretation of the specimen. We expand this case with further fluid preserved specimens, and demonstrate the importance of knowing if these specimens were collected by shooting or in other ways contain shotgun pellets e.g. introduced by previous unsuccessful hunting or by ingestion. Both from a scientific point of view and potentially also from a health point of view for collection staff and visiting scientists handling these specimens.

In the winter of 2022, a platypus specimen with the catalogue# NHMD-M01-28 was transported from the fluid preserved mammal collection at The Natural History Museum of Denmark, Copenhagen, to Aarhus University Hospital, Skejby, Denmark, with the intention of imaging the heart with MRI for the purpose of an ongoing study on cardiac muscle fibre architecture across mammalian hearts. This was performed outside clinical hours using a sequence previously optimized for cardiac tissue in ethanol preserved specimens with a field of view spanning most of the chest- and abdominal cavity. No abnormalities were noted by the operator of the scanner, who is not a trained veterinarian (author HL). Later, before the specimen was to be returned to the museum's collection, it was decided to subject it to another non-invasive imaging technique, CT, in order to digitize the entire specimen for future use and in particular generate a virtual dataset of bone morphology and bone density of the specimen. On this occasion, widely distributed centimetre sized hyperintense nodules were observed in the specimen by the scanner operators, none of whom are trained veterinarians (authors CCEP, KH and HL). After discussion with the collection manager and the mammal curator at The Natural History Museum of Denmark (authors DJK and PRM, respectively), it was decided to ask for advice from highly trained research and zoo veterinarians (authors CJAW, DS, AKOA and MFB) to form an opinion on what these hyperintense nodules could originate from e.g., a disease process in the preserved platypus.

Platypus NHMD-M01-28 was formally included in the mammal collection of The Natural History Museum of Denmark (at that time "Zoological Museum, University of Copenhagen")

on the 1<sup>st</sup> of July 1931. The specimen was obtained during "The Carlsberg Foundation's Oceanographical Expedition round the World 1928–30 under the Leadership of Professor Johannes Schmidt", also known as the fourth Dana expedition [16]. In Australia, the expedition was both collecting specimens and receiving donations from local scientists of floral and faunal samples of natural-historical value [17]. Although platypus were already at that time prohibited from being traded and/or exported from Australia, the then director of the Queensland Museum, Dr. Albert Heber Longman [18], was able to issue a license of export for a single preserved platypus. In this way platypus NHMD-M01-28, henceforth referred to as "the Dana platypus", came into possession of the expedition as a donation when visiting Brisbane between the 3<sup>rd</sup> and the 10<sup>th</sup> of March 1929 [17]. There is no record of the method of collection of the Dana platypus, but it was originally collected nearby Eidsvold, South East Queensland (25˚22' S, 151˚07' E) by Thomas Lane Bancroft and donated to the Queensland Museum (original catalogue# J2390) on the 21<sup>st</sup> of June 1915 (Janetzki, H., Queensland Museum, personal communication). There is no fixation or preservation record of the specimen, but it is presumed that since its inclusion in the collection of the Natural History Museum of Denmark it has been stored individually and according to custom in 70% v/v EtOH in a glass container for the last ~92 years until the described imaging event. There is no record of any re-fixation treatments (e.g. short 4% formaldehyde baths) of the specimen during this time.

After being offered several plausible pathological explanations for the dense nodules in the Dana platypus by veterinarians who based their evaluation on qualitative CT and MRI data, it was noted that the image intensity of nodules was potentially surpassing biological values. This could indicate that the nodules may in fact originate from some foreign objects introduced into the specimen before or after it was collected i.e. perimortem or postmortem. At this point it was decided to conduct a quantitative analysis to investigate the most likely origin of these nodules and if similar objects were present in other preserved platypus or other small fluid preserved mammals in the museum's collection. The presence of similar nodules in other platypus but not in other species would offer support to a pathological origin. This explanation, however, would be less likely if similar nodules were observed in other unrelated species, which instead would suggest an external origin e.g. shotgun pellets used for collecting the specimens, which later corroded to unrecognisable shapes and sizes after long-term storage in preservation fluids. Since the latter would be of relevance to researchers using fluid preserved museum specimens for comparative anatomy studies and curators and collection staff handling these specimens, we decided to dig into this unexpected mystery and investigate the origin of the dense nodules in the Dana platypus, and the potential effect that corroded shotgun lead pellets can have on quantitative imaging studies and the safety of handling affected specimens.

## Materials and methods

### Specimen information

Twenty six ethanol preserved and four pelts of small mammals in the collection of The Natural History Museum of Denmark collected or registered at the museum between 1849 and 1962 (Table 1) were included in the study. Since the study had its origin in an observation on the Dana platypus (*Ornithorhynchus anatinus* catalogue# NHMD-M01-28) all available platypus specimens in the collection were included except for stuffed specimens and dry bones. The inclusion of remaining species was determined by the collection era and the perceived likelihood of the specimen having been collected by the aid of a shotgun. Additionally, cadavers of six freshly culled laboratory rats (Sprague Dawley, 9 weeks of age, body mass = 238.3 ± 9.5 g, not used in other studies), all female, Forum breeding permit, (Aarhus University, breed under permission from the Danish Animal Experiments Inspectorate) were included to test

**Table 1. Fluid preserved mammal specimens in the collection of The Natural History Museum of Denmark (NHMD) having undergone x-ray examination for foreign objects.**

| Cat # | Binomial name | Common name | Type | Year | Nodules | Size (mm) | # |
|---|---|---|---|---|---|---|---|
| M07-CN958 | *Aplodontia rufa* | Mountain beaver | Fluid | 1893 | No | | |
| M03-CN195 | *Atopogale cubana* | Cuban solenodon | Fluid | 1886 | No | | |
| M07-CN398 | *Spermophilus citellus* | European ground squirrel | Fluid | 1885 | No | | |
| M02-CN462 | *Dactylopsila trivirgata* | Striped possum | Fluid | 1893 | No | | |
| M05-CN1074 | *Eidolon dupreanum* | Madagascan fruit bat | Fluid | 1894 | No | | |
| M07-CN1217 | *Gymnoromys roberti* | Voalavoanala | Fluid | 1895 | No | | |
| M10-CN1331 | *Mustela erminea* | Stoat | Fluid | 1909 | No | | |
| M01-3 | *Ornithorhynchus anatinus* | Platypus | Pelt | ukn. | No | | |
| M01-4 | *Ornithorhynchus anatinus* | Platypus | Pelt | 1879 | No | | |
| M01-11 | *Ornithorhynchus anatinus* | Platypus | Pelt | 1869 | No | | |
| M01-29 | *Ornithorhynchus anatinus* | Platypus | Pelt | 1936 | No | | |
| M01-25 | *Ornithorhynchus anatinus* | Platypus | Fluid | 1915 | Yes | 13.2 | 6 |
| M01-10 | *Ornithorhynchus anatinus* | Platypus | Fluid | 1865 | Yes | 11.7 | 18 |
| M01-24 | *Ornithorhynchus anatinus* | Platypus | Fluid | 1917 | Yes | 8.4 | 5 |
| M01-28 | *Ornithorhynchus anatinus* | Platypus | Fluid | 1915 | Yes | 10.4 | 21 |
| M01-34 | *Ornithorhynchus anatinus* | Platypus | Fluid | 1922 | Yes | 5.0 | 12 |
| M01-35 | *Ornithorhynchus anatinus* | Platypus | Fluid | 1922 | No | | |
| M01-9 | *Ornithorhynchus anatinus* | Platypus | Fluid | 1865 | Yes | 6.4 | 17 |
| Ucat | *Ornithorhynchus anatinus* | Platypus | Fluid | 1950 | No | | |
| M02-CN299 | *Perameles gunnii* | Eastern barred bandicoot | Fluid | 1914 | No | | |
| M02-CN439 | *Petaurus breviceps* | Sugar glider | Fluid | 1931 | No | | |
| M02-CN297 | *Phascogale tapoatafa* | Brush-tailed phascogale | Fluid | 1902 | No | | |
| M02-CN265 | *Philander opossum* | Gray four-eyed opossum | Fluid | 1893 | No | | |
| M02-CN440 | *Pseudocheirus peregrinus* | Common ringtail possum | Fluid | 1931 | Yes | 7.5 | 8 |
| M05-CN2901 | *Pteropus rayneri* | Solomons flying fox | Fluid | 1962 | Yes | 5.2 | 9 |
| M07-CN13 | *Ratufa macroura* | Grizzled giant squirrel | Fluid | 1865 | No | | |
| M07-CN1164 | *Callosciurus finlaysonii* | Finlayson's squirrel | Fluid | 1900 | Yes | 6.6 | 5 |
| M07-CN1163 | *Callosciurus finlaysonii* | Finlayson's squirrel | Fluid | 1900 | Yes | 6.8 | 4 |
| M07-CN17 | *Sciurus colliaei* | Collie's squirrel | Fluid | 1849 | No | | |
| M06-CN256 | *Tupaia belangeri* | Northern treeshrew | Fluid | 1893 | No | | |

Nodule size in mm, number (#) refer to number of large distinct nodules in the specimen.

the effect of lead pellets (Danish shot size 5 = 3 mm in diameter, Gyttorp Cartridge Company AB, Nora, Sweden) on measurements of bone mineral content using quantitative CT imaging. This study involved only museum specimens and cadavers of previously culled rats not used in any experiments and did not require an animal experimental permit.

## Digital x-ray imaging

All specimens but the Dana platypus were imaged using projection based digital x-ray imaging using a XPERT 80-L system (KUBTEC Scientific, Stratford, Connecticut, USA) with x-ray tube voltage = 70–125 kV and x-ray tube current = 42–110 μA.

## X-ray computed tomography imaging

The Dana platypus and the six laboratory rat cadavers were imaged with x-ray computed tomography imaging using a Canon Aquilion Prime SP system with the following parameters:

x-ray tube voltage = 80, 100, 120, 140 kVp, x-ray tube current = 260 mA, integration time = 1000 ms, field-of-view = 256 × 256 × 756 mm$^3$, spatial resolution = 0.5 mm isotropic, convolution kernel = FC18, acquisition time = 60 s pr. scan. Underneath the specimens, a Mindways QCT Pro bone mineral calibration phantom was positioned to calibrate x-ray attenuation values to bone mineral density (mg/mm$^3$ equivalent aqueous $K_2HPO_4$).

Additionally, the anterior portion (head and shoulder girdle) of the Dana platypus and excised nodules from the back and the right hind limb were imaged at a higher resolution using a Medical XtremeCT system (Scanco, Brüttisellen, Switzerland) with the following parameters: x-ray tube voltage = 59.4 kVp, x-ray tube current = 119 µA, integration time = 132 ms, field-of-view = 70 × 70 × 150 mm$^3$, spatial resolution = 0.082 mm isotropic, acquisition time = 1.5 h pr. scan.

Finally, the neurocranium and part of the bill of the Dana platypus was imaged using a Uni-TOM XL Spectral system (TESCAN GROUP, Brno, Czech Republic) equipped with a hyper-spectral detector (channel size of 1 keV) with the following parameters: x-ray tube voltage = 160 kVp, x-ray tube current = 94 µA, x-ray tube power = 20 W, integration time = 83 ms, field-of-view = 76.4 × 76.4 × 27.1 mm$^3$, spatial resolution = 0.108197 mm, acquisition time = ~10 h. K-edge subtraction to highlight the Pb signal (K-edge at 88.0045 keV) was performed using 1 channels width and 3 channels separation (91 keV– 85 keV).

Beef samples used to test the corrosion of lead pellets in tissue (see description below) were imaged using the same UniTOM XL Spectral system as described above, but using its standard integrating detector and using the following parameters: x-ray tube voltage = 80 kVp, x-ray tube current = 625 µA, x-ray tube power = 50 W, integration time = 25 ms, field-of-view = 48 × 48 × 38.35 mm$^3$, spatial resolution = 0.05 mm isotropic, acquisition time = 12 min per sample.

## Magnetic resonance imaging

The Dana platypus was imaged with magnetic resonance imaging using a Siemens Magnetom Skyra 3 T system equipped with a 15 channel transmit/receive knee coil using a $T_2$-weighted 3D spin echo sequence with the following parameters: repetition time = 1000 ms, echo time = 132 ms, refocusing flip angle = 120˚, field-of-view = 196 × 196 × 108.8 mm$^3$, spatial resolution 0.34 mm isotropic, number of averages = 4, acquisition time = 2 h.

## Image analysis

Image J (version 1.50e) and OsiriX DICOM Viewer were used for image viewing and reslicing. Bone mineral content analysis was conducted as previously described [11, 19].

## Chemical analysis of foreign objects

The nodules in the Dana platypus were determined to be foreign objects (FO) based on the hyperintense appearance on CT images and all 21 major objects were numbered in the order they appeared from the cranial end of the specimen. Foreign object 11 (dorso-lateral mid thorax), 12 (in the abdomen), 14 (caudal abdomen), 16 (right hind limb, lateral to the tibia) and 21 (lateral tail) were excised and used for chemical analysis (FO 11, 12, 14, 21) and high-resolution imaging (FO 11, 16). Excision cuts in the specimen were carefully sutured to keep the integrity of the specimen.

A flame test was conducted on the entire FO 11 as well as on a slurry of the core and crust, respectively, of FO 12 in double distilled water. The flame test was conducted accordingly to the Royal Society of Chemistry (United Kingdom) guidelines using wooden splints and double distilled water as solvent [20]. As a reference of flame colours of common biometals, calcium

(Ca), potassium (K), sodium (Na) and iron (Fe), salt solutions of $CaCl_2$, KCl, NaCl at a concentration of 1 M as well as a saturated slurry of $Fe_2O_3$ were used.

To test for traces of common metals in shotgun lead pellet alloys, lead (Pb), antimony (Sb), tin (Sn) and arsenic (As), a range of test papers and kits were used: Pb, test paper 1, Water-strips, Joygain, Hong Kong, SAR China; Pb, test paper 2, Plumbtesmo, Macherey-Nagel, Düren, Germany; Sb, Antimon-Testpapier, Macherey-Nagel, Düren, Germany; Sn, Quantofix Tin, Macherey-Nagel, Düren, Germany; As, Arsenic Test, MQuant, Supelco/Merck, Darmstadt, Germany. The sensitivity of the two used Pb test papers was validated using a serial dilution of a saturated solution of $PbCl_2$ in double distilled water. Test paper 2 proved most sensitive with a sensitivity of 9.7 mg/l $PbCl_2$, i.e. 34.9 μM $Pb^{2+}$. Excised foreign objects were homogenized using a Benchmark BeadBug 6 microtube homogeniser operating at 3500 rpm for 10 cycles (30 s shaking / 30 s rest) and equipped with 2 ml microtubes containing five 2.8 mm stainless steel beads, 1 ml ultrapure water and ~300 mg sample. Additionally, shotgun lead pellets (Danish shot size 5 = 3 mm in diameter, Gyttorp Cartridge Company AB, Nora, Sweden) were used for reference measurements of metal alloys. These shot pellets were treated in the same way as the samples of excised foreign objects.

Solubility of the whitish core of FO 14 at different temperatures was measured by first incubating the homogenate in double distilled water at 90˚C on a Benchmark Multi-Therm thermoshaker at 500 rpm for 2 h. Then the solution was filtered for particles using a strainer with a 100 μm mesh size. The solution was then cooled in steps to 80, 60, 40 and 20˚C without shaking to let the excess of the dissolved salt precipitate as the temperature was lowered and left to stabilize for 2 h for each temperature. At each temperature step, a dilution series of 1, 2, 4, 8, 16, 32, 64, 128 and 256x was prepared for the salt solution, and the concentration of Pb was measured according to the calibration dilution series of $PbCl_2$ on both Pb test paper 1 and 2.

### Chemical analysis of preservation fluids

Preservation fluids of specimens abbreviated Dtri CN462, Edup CN1074, Merm CN1331, Oana M25, Oana M9, Oana M10, Oana M24, Oana M28, Oana M34-35 (stored in same container), Oana Ucat, Pper CN440, Pray CN2901, Cfin CN1163-1164 (stored in same container), Scol CN17 and Acub CN195 (see Table 1 for full specimen information) were tested for residues of the toxic metals Pb and As. First, 40 ml samples were distilled at 79˚C to evaporate the ethanol content of the samples. Distillation was continued until each sample was concentrated by 30x. Many concentrated samples consisted of a lipid top layer and a watery bottom lay, thus the presence of Pb in either the lipid or the water phase was tested by pipetting 10 μl of each phase onto Pb test paper 2. Subsequently, concentrated samples were diluted to 3.5x initial concentration by adding double distilled water to allow for a sufficient volume to conduct As-tests.

### Corrosion test of lead pellets

A corrosion test of lead shot pellets (Danish shot size 5 = 3 mm diameter, Gyttorp Cartridge Company AB, Nora, Sweden) stored in either double distilled water (pH = 5.01), 70% v/v EtOH with remainder being double distilled water (pH = 6.03, but with the uncertainty of measuring acidity in EtOH/water-solutions [21], 70% v/v EtOH with remainder being phosphate buffered saline (pH = 8.61), 100% EtOH, 4% v/v phosphate buffered formaldehyde (pH = 6.84), 4% v/v unbuffered formaldehyde (pH = 3.97) or 40% v/v unbuffered formaldehyde (pH = 3.18) was conducted as previously described [22]. In short, lead pellets were first freed of any corrosion products or grease by shaking at 2500 rpm for three cycles (30 s shaking / 30 s rest) on a Benchmark BeadBug 6 microtube homogenizor equipped with 2 ml microtubes containing 0.5 mm silica glass beads. This was performed first in dilute hydrochloric

acid (3 M) and then in 100% acetone. Each pellet was weighed with 0.1 mg precision, and for each of the seven test solutions six pellets were placed in individual 1.5 ml Eppendorf tubes containing 1 ml of the test solution. Tubes were left in a horizontal position on a laboratory rocker for 22 days. After that time, corrosion products were removed again as described above and pellets were reweighed. Corrosion rate was calculated as the mass in µg of lost material per mm$^2$ surface area of the spherical pellets per day. Following the corrosion test, test solutions were centrifuged at 2500 g for 5 min, and supernatants were tested for the presence of dissolved Pb, Sb, Sn and As using test papers and kits as described above.

To investigate the corrosion of lead shot pellets within tissue samples stored in various preservation fluids, six $2 \times 2 \times 2$ cm$^3$ pieces of beef sirloin were prepared and a single lead shot pellet (Danish shot size 5 = 3 mm diameter, Gyttorp Cartridge Company AB, Nora, Sweden) was inserted into the centre of the beef cube using a forceps. The beef samples were preserved for a duration of 256 days (~8.5 months) with gentle rocking in 35 ml of either 70% v/v EtOH (with remainder being double distilled water), 4% v/v phosphate buffered formaldehyde, 4% v/v unbuffered formaldehyde, 40% v/v unbuffered formaldehyde, or in either 4% v/v phosphate buffered formaldehyde or 4% v/v unbuffered formaldehyde for 7 days and then in 70% v/v EtOH to mimic a typical fixation/preservation procedure. After the preservation period, lead pellets were carefully removed using forceps with minimum disruption of tissue, and the samples were imaged using the UniTOM XL Spectral system as described above. The volume of corrosion products remaining in the beef samples was quantified by counting the number of voxels with signal intensities surpassing those of unaffected beef tissue (>12800).

## Statistical analysis

Data is shown throughout as mean values ± standard deviations. Analyses of significant differences between two groups were performed using two-tailed Student's t-test (paired or unpaired as relevant), and for more than two groups using ANOVA. For repeated measures, one-way ANOVA with repeated measures was performed with Tukey's honest significance test for post hoc test of significant differences between groups. We considered p-values less than 0.05 to signify statistical significance.

## Results

### First observations on the Dana platypus and suggested pathologies

Superficially, the Dana platypus presented with no obvious injuries, except from a cavity in the soft upper bill at the level of the right premaxilla (Fig 1A). Magnetic resonance imaging focused at the cardiac region did not immediately reveal any abnormalities, however, subsequent CT imaging showed widely distributed centimetre sized hyperintense nodules (Fig 1B). Due to their large size and nodular appearance, these objects were initially disregarded as potential projectiles and their apparently random distribution also suggested that they had not been inserted for the purpose of mounting the specimen in a certain position. After a qualitative evaluation (i.e. not focusing on x-ray attenuation levels in nodules), it was decided that the nodules could be of neoplastic or infectious nature. Presenting this specific mind-set (i.e. potentially inducing preconceived bias), advice was requested of experienced research and zoo veterinarians. Based on limited and only qualitative imaging material (S1 File) the notion that nodules could be neoplastic was supported and several additional explanations were suggested e.g. mucormycosis with mineralisation or tuberculosis (Table 2). However, the scans were not considered pathognomonic for any of these conditions. Subsequently, subdermal and abdominal nodules were excised from the Dana platypus for further analysis. High resolution micro-CT imaging and the physical sectioning of two nodules revealed their consistence of a

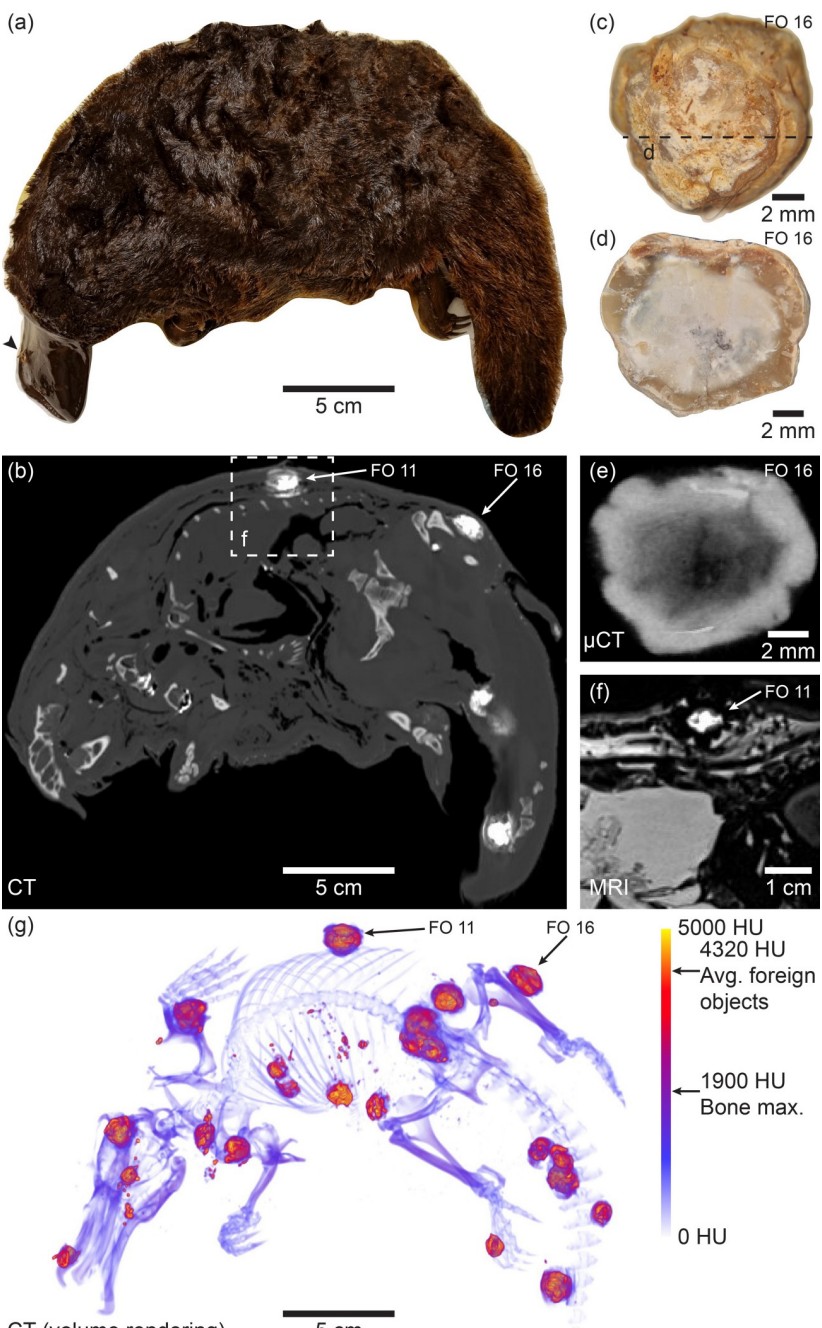

**Fig 1. The Dana platypus (NHMD-M01-28) collected during The Carlsberg Foundation's Oceanographical Expedition round the World 1928–30.** (a) Apart from a single puncture at the right side of the bill (black arrowhead) the specimen shows no superficial evidence of collection method; (b) Virtual coronal section from x-ray computed tomography (CT) showing hyperdense foreign objects (FO) embedded in the specimen; (c) Foreign object 16 dissected free from the thigh musculature appears as a centimetre sized nodule with great visual resemblance to dried chewing gum, but more brittle in nature; (d) Slice through FO 16 showing a whitish core and a brownish crust; (e) Virtual section of FO 16 from micro x-ray computed tomography (μCT) showing the crust is more radiodense than the core; (f) Virtual coronal section (similar to (b) from magnetic resonance imaging (MRI) of FO 11. In contrast to bony material that appears hypodense in the specific T2-weighted spin echo sequence, the foreign object appears hyperdense, but with a low level of image artifacts. This indicates that the foreign object does not consist of either bone mineral or paramagnetic substances; (g) Volume rendering from CT of the Dana platypus allowing for the appreciation of the large number of foreign objects. At least 21 centimetre sized objects can be observed. The volume rendering is calibrated to a 0–5000 Hounsfield unit (HU) calibration bar to demonstrate radiodensity of the skeleton and foreign objects. Skeletal components rarely exceed 1900 HU, but the average radiodensity of the foreign objects is 4320 HU indicating a non-biological origin.

**Table 2. Suggested pathologies in Dana platypus based on limited imaging data.**

| Pathology | Observations speaking for | Observations speaking against |
|---|---|---|
| Osteosarcoma with secondary metastases | Proliferative mineralised lesions, 'cannonball' appearance | No obvious destructive process within bony tissue. No obvious primary lesion, and secondaries often pulmonary in clinical cases |
| Other bone forming neoplastic lesions | Multiple mineralised lesions | Round and concentrated rather than diffuse appearance |
| Mucormycosis | Well known disease in this species, causing widespread and potentially extensive nodular lesions | Uncharacteristic for lesions to be mineralised. |
| Tuberculosis | Commonly found in many different species. Lesions can become calcified. | Typically processes are concentrated in single organs, such as the lungs. |
| Tumoral calcinosis | Multiple calcified nodules. | Only described in humans |
| Tophaceous gout | Multiple radiodense nodules. Observed in several species. | Typically smaller nodules. |

radiodense and physically dense outer crust and a softer core (Fig 1C–1E). A reinspection of MR images revealed that nodules were in fact observable (Fig 1F), and, in opposition to bone material presenting hypo intense using the specific $T_2$-weighted spin echo sequence, the nodule core presented hyperintense compared to surrounding tissue. Also, nodules did not cause pronounced image artifacts, suggesting the absence of paramagnetic materials such as iron (Fig 1F). At this point, a quantitative analysis of x-ray attenuation in nodules was conducted revealing CT values of 4320 ± 531 Hounsfield units (HU) (Fig 1G). This is markedly higher than most dense bony structures found in humans of approximately 1900 HU [23, 24], and thus a careful investigation for a non-biological origin of the nodules, or more precisely the foreign objects, was warranted.

## Chemical analysis of nodules

The apparently random distribution of nodules suggested that their origin were more likely that of corroded projectiles e.g. shotgun lead pellets than that of other types of foreign objects introduced to preserve or mount the specimen. Thus, chemical analysis was conducted to test the hypothesis that nodules were corroded lead pellets. A flame test of both entire excised nodules as well as core and crust material prepared separately as a slurry in double distilled water, revealed flame colours in pink, light blue or whitish nuances which is not characteristic for some of the most abundant biometals: Ca, K, Na and Fe, all with distinct flame colours (Fig 2A). Two different brands of test papers specifically prepared to detect Pb was validated and calibrated against a serial dilution of $PbCl_2$ in double distilled water (Fig 2B and 2C), and additional test papers and kits to detect Sb, Sn and As, revealed the presence of all four metals in the excised nodules (Fig 2B–2D). Since neither of these metals are usually found in high quantities in healthy biological tissue, this strongly suggested that the nodules were foreign objects and very likely corroded shotgun pellets. To inform whether lead salts of the corroded lead pellets were reaction products of metallic lead and fixation or preservation chemicals, e.g. formaldehyde or EtOH, the solubility of core and crust salts of foreign object14 was tested at 20, 40, 60 and 80°C. This showed that remaining lead salts were highly insoluble compared to other relevant lead salts such as lead(II) acetate, lead(II) formate and lead(II) chloride (reference values from [25]) even at high temperatures (Fig 2E and 2F).

## X-ray spectral and hyperspectral analysis of nodules

Different elements have different x-ray attenuation signatures in function of the x-ray photon energy. Thus, to expand on the chemical analysis suggesting a non-biological origin of the nodules, an x-ray spectral analysis was conducted on six of the nodules of the Dana platypus

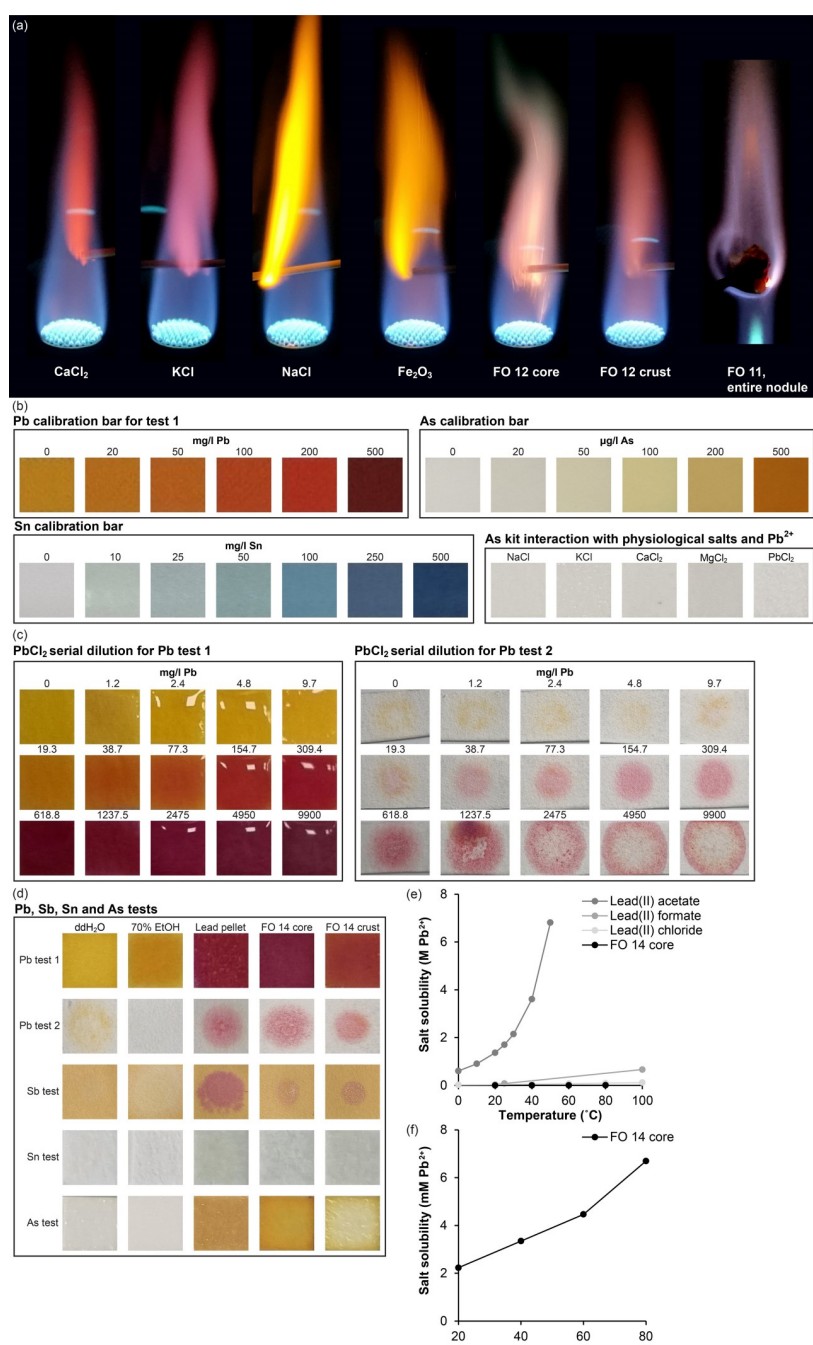

**Fig 2. Chemical tests to reveal elemental components of foreign objects. (a)** Flame test of metal salts of most common biometals and pulverised core and crust of foreign object (FO) 12 and the whole of FO 11; **(b)** Calibration bar for lead (Pb) test paper 1, Tin (Sn) test paper, and arsenic (As) test kit. Additionally, test for interactions between As test kit and common physiological salts and Pb$^{2+}$. No interactions were observed; **(c)** Serial dilution of PbCl$_2$ on Pb test paper 1 and 2 to reveal lower detection limit of Pb on these test papers. Test paper 2 was most sensitive with a detection limit of 9.7 mg/l of PbCl$_2$ equalling 34.9 µM Pb$^{2+}$; **(d)** Analysis for the presence of Pb, antimony (Sb), Sn and As in double distilled water, 70% ethanol (EtOH), homogenized lead shot pellet, and homogenized core and crust of FO 14. No interaction was observed for any metal for water and ethanol whereas all four metals were detected in the lead shot pellet and in both core and crust of FO 14; **(e)** Salt solubility of FO 14 core material as a function of temperature in comparison with different lead salts (literature values); **(f)** Magnification of the graph above with a focus on FO 14 core solubility. The solubility of the FO 14 core material is much lower than would be expected for reaction products between ethanol and formaldehyde.

and on lead shotgun pellets (Fig 3A). Conveniently within the energy range of most conventional clinical CT scanners, 80–140 keVp, Pb has a K-edge at 88 kV, resulting in a steep increase in x-ray absorption at that energy level which is evident in the idealised spectrum of monochromatic x-rays (Fig 3B) (reference values from [26]). On the other hand, naturally occurring minerals such as Ca (e.g. in bones), does not have any K-edges in this range, and thus for bony materials, the x-ray attenuation would be expected to only decrease at increasing x-ray energies as displayed in Fig 3B. A comparable bone signature was found at regions of interest positioned at skeletal sites of the left tibia, skull and a thoracic vertebra of the Dana platypus (Fig 3A and 3C) analysed with the polychromatic x-ray source of the clinical CT scanner when setup to scan the specimen at individual energies. On the other hand, regions of interest placed within the foreign objects showed a markedly different signature with an initial decrease in x-ray attenuation from 80 to 100 keVp, followed by an increase in attenuation at higher energies (Fig 3A and 3D). The attenuation signature of lead shot pellets was most similar to the idealised monochromatic spectrum of Pb with a steep increase in attenuation between 80 and 100 keVp, although the subsequent decrease in attenuation at higher energies was not observed for lead shot pellets with the polychromatic x-ray source and conventional integrating detector of the used clinical CT scanner (Fig 3A and 3E). To elaborate, we performed hyperspectral CT imaging with a dedicated micro-CT system on the neurocranium of the Dana platypus (Fig 3F), since foreign objects were also contained within this region. In contrast to the insensitivity of the clinical CT system to detect K-edges within the foreign objects, the hyperspectral detector of the micro-CT system clearly detected a K-edge at 88 keV of regions of interest placed within the foreign objects, which was not observed within bone voxels (Fig 3G and 3H). This allowed for K-edge subtraction (integrating and subtracting channels on either side of a K-edge to produce element specific images) and the precise mapping of Pb in the neurocranium (Fig 3F middle and right panels). In addition to confirming the presence of large concentrations of Pb in the foreign objects within the Dana platypus, hyperspectral imaging also showed that the widely distributed small dense particles within the fur of the specimen were not corroded lead residues (no Pb K-edge in the particle cloud surrounding the sample in Fig 3F lower right panel).

## Shot trajectory

In the Dana platypus, one large foreign object was positioned in the posterior portion of the right cerebral hemisphere (Figs 1G and 3A ("FO 3"), 3f and 4a, 4b). Since the neurocranium is a nearly closed bony structure, an encapsulated foreign object would most likely leave a trace of entry and thus carry information about the shot trajectory. A micro-CT scan of the head region revealed bone fragments and hyperintense fragments or smears in the skull (Figs 3F and 4A and 4B). The position of fragments and the large encapsulated foreign object suggested a shot trajectory with a vertical component close to parallel or just a few degrees lower than the bill and a horizontal component approximately 30˚ to the right of facing the animal directly (Fig 4B). Although entry wounds were not immediately visible in the wet specimen and had been completely overlooked during the initial handling of the specimen, after 10 minutes of drying when most EtOH at the surface was evaporated, an entry wound was clearly visible on the leftmost portion of the leathery upper bill flap (Fig 4C, right) at the position where it would be expected from the micro-CT determined shot trajectory. Likewise, a small nearly sealed entry wound just left of the midline was observed at the lower bill after the preservation fluid had evaporated from the surface (Fig 4C, lower left). The position of this entry wound was consistent with the interpreted shot trajectory and the obvious cavity on the right side of the upper bill being an exit wound (Fig 4C).

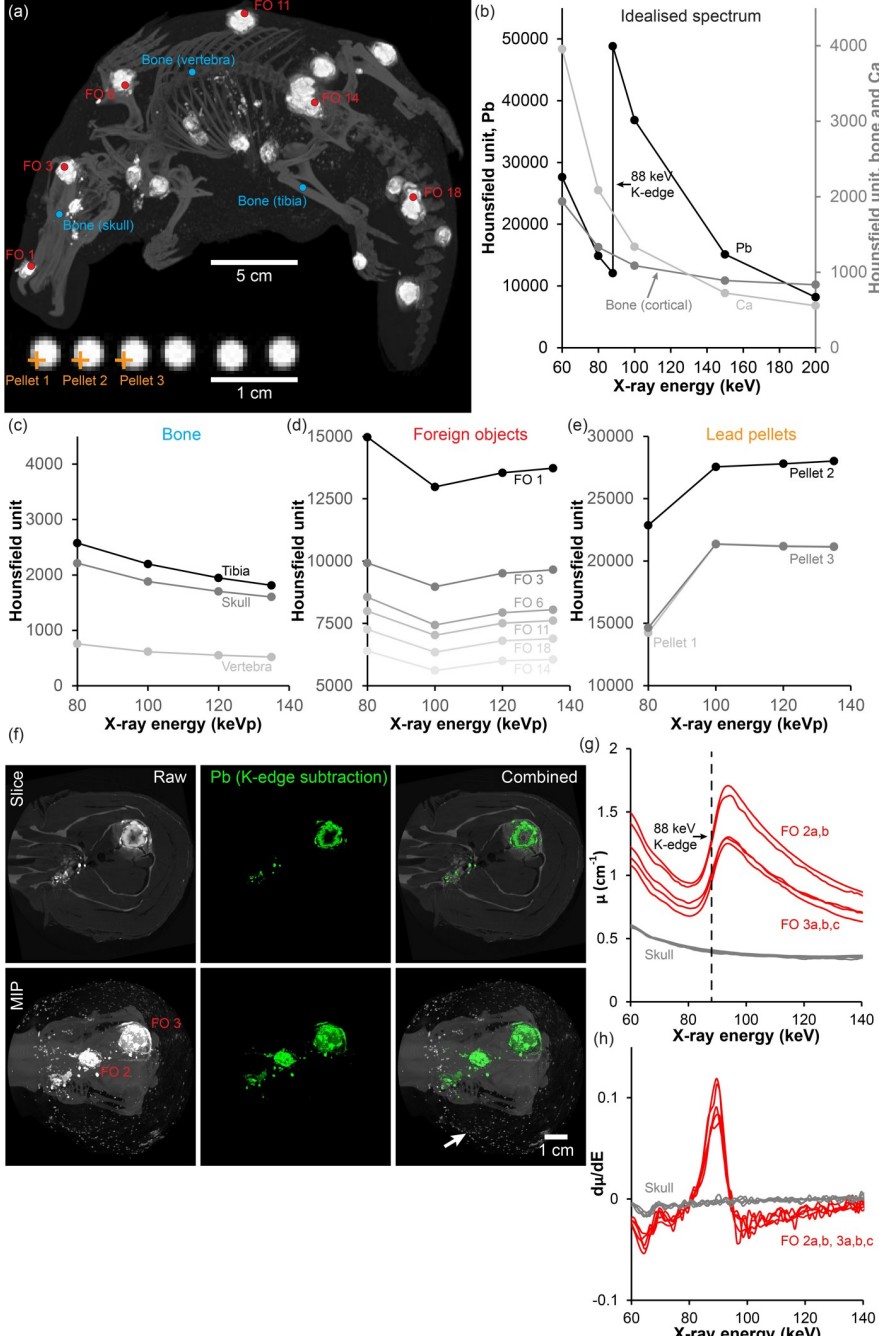

**Fig 3. X-ray spectral analysis of foreign objects in the Dana platypus and in lead pellets. (a)** Overview of the Dana platypus and magnified lead pellets showing regions (circles) or points (crosshair) of interest; **(b)** Idealised x-ray attenuation spectra for lead (Pb) (primary second axis), cortical bone and calcium (Ca) (both on secondary second axis) at monochromatic x-ray photon energies from 60–200 keV. Lead has a sharp K-edge at 88 keV allowing for material identification in this range; **(c)** Polychromatic x-ray computed tomography (CT) attenuation spectra for peak energies between 80 and 135 keVp for the Dana platypus bone at regions of interest placed at the tibia, skull and a vertebra, respectively. All three spectra follow the expected decrease in attenuation at higher energies as predicted by the idealised spectrum for bone (b); **(d)** Polychromatic CT attenuation spectra for six foreign objects (FO) in the Dana platypus. The spectra are distinct from the bone spectra (c) showing an increase in attenuation at higher peak energies, but they are also not similar to the idealised lead spectrum (b) probably due to the polychromatic nature of conventional CT imaging; **(e)** Polychromatic CT attenuation spectra for the peripheral partial volume effected region of three lead pellets. In spite of the partial volume effect (necessary to avoid the Hounsfield unit threshold) and the polychromatic nature of conventional CT imaging, the lead pellet spectra are comparable to the idealised spectrum of

lead (b) showing a pronounced increase in attenuation between 80 and 100 kVp; (**f**) Hyperspectral CT imaging of the neurocranium of the Dana platypus presented as coronal slices (top row) and maximum intensity projections (MIP, bottom row) of an averaged image of the 20–120 keV energy channels. K-edge subtraction at the K-edge of lead (88 keV) allows for material specific mapping of lead within the CT volume; (**g**) X-ray attenuation spectra (linear attenuation coefficient, μ, over x-ray energy) for five regions placed within FO 2 and FO 3 and five regions with skull bone. The sharp K-edge at 88 keV in the foreign objects confirms the presence of lead in foreign objects; (**h**) Derived curves of attenuation spectra in (g) showing maximum increase in attenuation at 88 keV.

## Foreign objects in other fluid preserved mammals

Although chemical analysis and x-ray spectral analysis of nodules in the Dana platypus and the discovery of two entry wounds and bone fragments with a consistent shot trajectory all strongly suggested the nodules to be foreign objects, very likely corroded lead shot pellets where tissue infiltration by corrosion products made them much larger than the original pellets, more support for this hypothesis would be gained if similar nodules were found in other unrelated mammals collected in other locations. Therefore, another 29 mammal specimens (25 fluid preserved and four dry pelts) were imaged with two-dimensional digital x-ray imaging (Table 1). This included the entire collection of platypus at the Natural History Museum of Denmark totalling eight fluid preserved platypus (including the Dana platypus), six of which contained hyperintense nodules (Fig 5A), and four pelts without any nodules. In addition, hyperintense nodules were found in an Australian opossum (*Pseudocheirus peregrinus* catalog# NHMD-M02-CN440), two Southeast Asian squirrels (*Callosciurus finlaysonii* NHMD-M07-CN1163 and M07-CN1164) and a Solomon Islands flying fox (*Pteropus rayneri* M05-NHMD-CN2901) (Fig 5A and Table 1). The close resemblance of nodules in these marsupial and placental mammals to the monotreme platypus, suggested a shared origin as foreign objects rather than a pathological nature, and very likely that of shot pellets. The *Pteropus rayneri* NHMD-CN2901 specimen was collected during the Danish Noona Dan Expedition to the Pacific in 1961–62, and the shotgun and its custom-made adapter sleeve which were most likely used for the collection of the flying fox was until 2023 in the weapons collection at the Natural History Museum of Denmark but was recently transferred to the Danish National Museum (Fig 5B).

## Chemical analysis of Pb and As in preservation fluids

Both Pb and As are highly toxic metals, and since both metals are common components of lead shot alloys and both were detected in the chemically analysed nodules of the Dana platypus, it was of interest to test the presence of these metals in the preservation fluid of the Dana platypus and the like of the several other fluid preserved specimens found to contain corroded shot pellets (n = 9) and compare with preservations fluids of specimens that were seemingly lead pellet free (n = 6). Fluid samples were heated to 79°C to ensure evaporation of ethanol and concentrated to 30x to measure residual Pb. Several concentrated samples consisted of a lipid upper phase and watery bottom phase, and both phases were tested on the most sensitive brand of Pb test paper. Lead was not detected in either phase of the concentrated samples meaning that original preservation fluids contained < 1.2 μM $Pb^{2+}$ (Fig 6A). To produce a sufficient volume to run the applied As test kit, concentrated fluid samples were diluted with double distilled water to reach a 3.5x concentration of the original preservation fluid. In contrast to the negative Pb test results, all but two samples tested positive for the presence of As with a peak concentration of 114 μg/l (1.5 μM As) (Fig 6B and 6C). Although both of the two preservation fluid samples without traces of As were found within the specimens seemingly without shot pellets, many other preservation fluids of specimens without pellets still contained traces of As (Fig 6B). There was no significant difference between As concentration in preservation

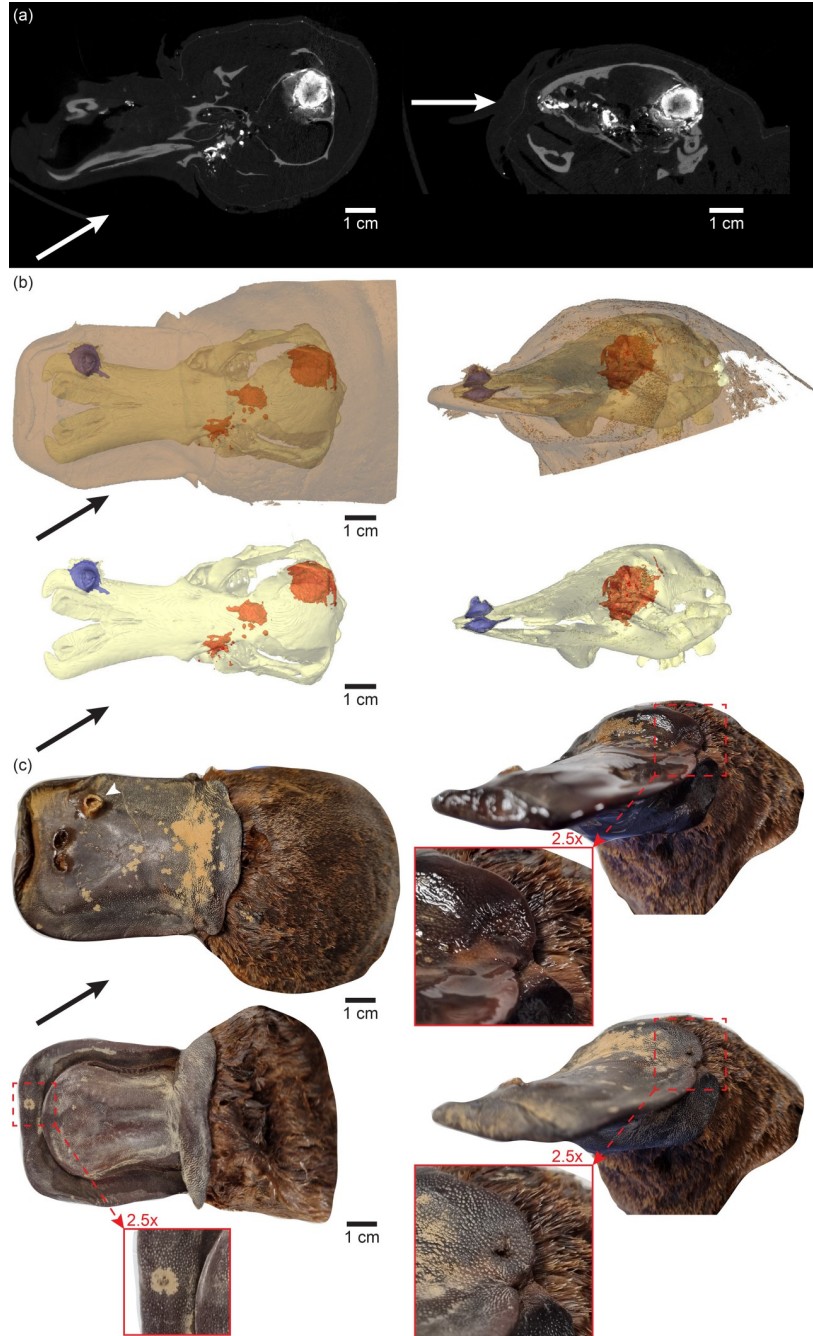

**Fig 4. Shot trajectory. (a)** Coronal (left) and oblique sagittal (right) virtual micro-CT sections in the head of the Dana platypus reveal a large foreign object in the right cerebral hemisphere and lead fragments and/or smears show the most likely shot trajectory; **(b)** Three-dimensional reconstructions of the head in a dorsal view (left) and viewed along the probable shot trajectory (right). Red and blue structures are hyperintense foreign objects; **(c)** Photos of the Dana platypus from the same viewpoints as in (b) and from a ventral viewpoint in addition (lower left). White arrowhead point to the exit wound on the dorsal portion of the bill. Magnifications show entry wounds. In the 70% ethanol soaked specimen (upper right) it is easy to overlook the small entry wound, whereas it is obvious after superficial preservation fluid has evaporated off (lower left and lower right).

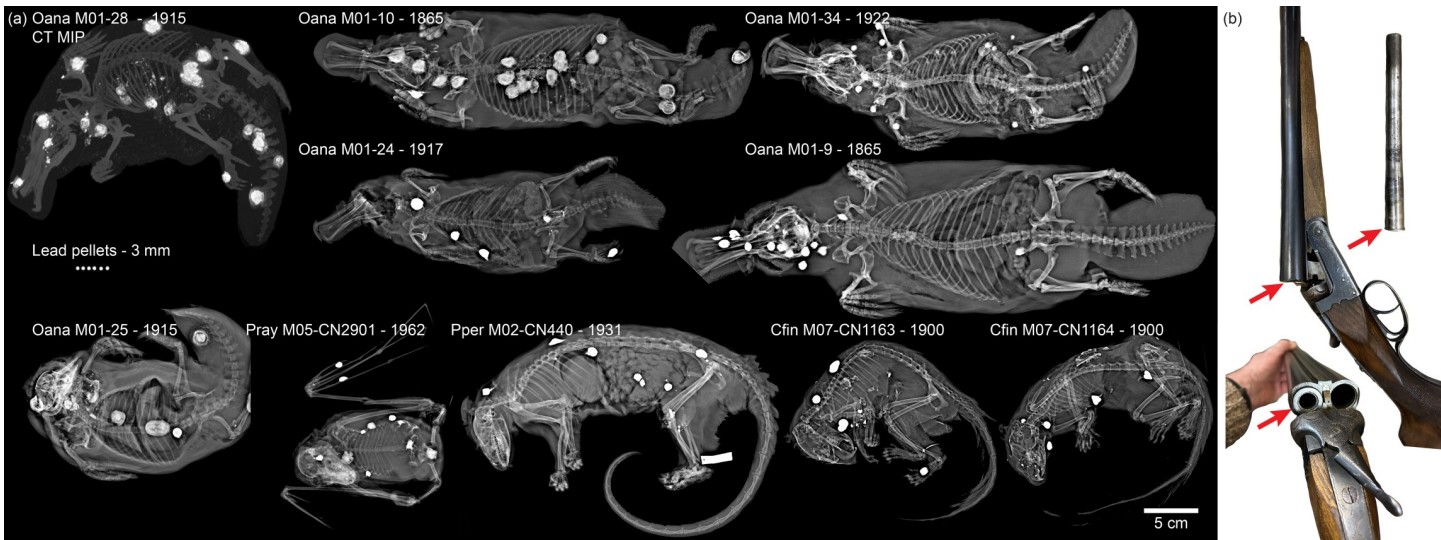

**Fig 5. Museum specimens collected with the shotgun. (a)** X-ray overview of small mammals in the fluid collection of The Natural History Museum of Denmark containing shotgun pellets. Oana = *Ornithorhynchus anatinus*, Pray = *Pteropus rayneri*, Pper = *Pseudocheirus peregrinus*, Cfin = *Callosciurus finlaysonii*. Six 3 mm lead pellets are shown (middle to the left) for comparison of size with foreign objects in the specimens; **(b)** Historical shotgun previously housed in the weapons collection of the The Natural History Museum of Denmark. This weapon and the custom made adapter sleeve (red arrow) was used at the Danish Noona Dan Expedition to the Pacific in 1961–62 and the Solomon Islands flying fox (Pray M05-CN2901) was most likely collected using this weapon.

fluids containing specimens with or without visible foreign objects (unpaired t-test, n = 6 and n = 9, p = 0.29) (Fig 6C).

## Corrosion rate of lead shot pellets in different fluids

To test the corrosion rate of lead shot pellets in different fluids with relevance to fluid preservation, lead pellets were stored for 22 days in either double distilled water, 70% EtOH (diluted from 100% with double distilled water), 70% EtOH (diluted from 100% with phosphate buffered physiological saline solution), 100% EtOH, 4% phosphate buffered formaldehyde, 4% unbuffered formaldehyde and saturated 40% unbuffered formaldehyde. After the storage period, the different fluids were tested for the presence of Pb, Sb, Sn and As. All but the 70% EtOH (PBS), 100% EtOH and the 4% buffered formaldehyde solution tested positive for the presence of Pb, although only barely for the 70% EtOH sample (Fig 7A). All solutions apart from the 100% EtOH solution tested positive for Sb (Fig 7A). Tin was only detected with certainty in the 40% unbuffered formaldehyde solution (Fig 7A). Arsenic was detected at low concentrations in the double distilled water and the 4% buffered formaldehyde solutions and at a high concentration in the 70% EtOH (PBS) solution (Fig 7A).

Corrosion rate, measured as the mass of lost pellet material per surface area per day, was highest when storing shot pellets in 4% unbuffered formaldehyde, followed by double distilled water and 40% unbuffered formaldehyde with significantly lower corrosion rates in 70% EtOH, 70% EtOH (PBS), 100% EtOH and 4% buffered formaldehyde (Fig 7B).

Based on corrosion rate, the extrapolated time it would take to completely corrode shot pellets of the given size (3 mm in diameter) and mass (165.4 ± 8.4 mg) was calculated. This ranged from 5.1 years in 4% unbuffered formaldehyde to 64.6 years in 70% EtOH (Fig 7C).

Preserving beef samples containing a single lead pellet in various preservation fluids also demonstrated differences in the formation of corrosion products over the course of 8.5 months (Fig 7D and 7E). Whereas preservation directly in 70% EtOH or initial fixation in buffered 4% formaldehyde followed by storage in 70% EtOH, which can be considered normal practice for

(a)

**Pb in lipid (L) and water (W) fractions of 30x concentrated preservation fluids of specimens with (+) and without (-) nodules**

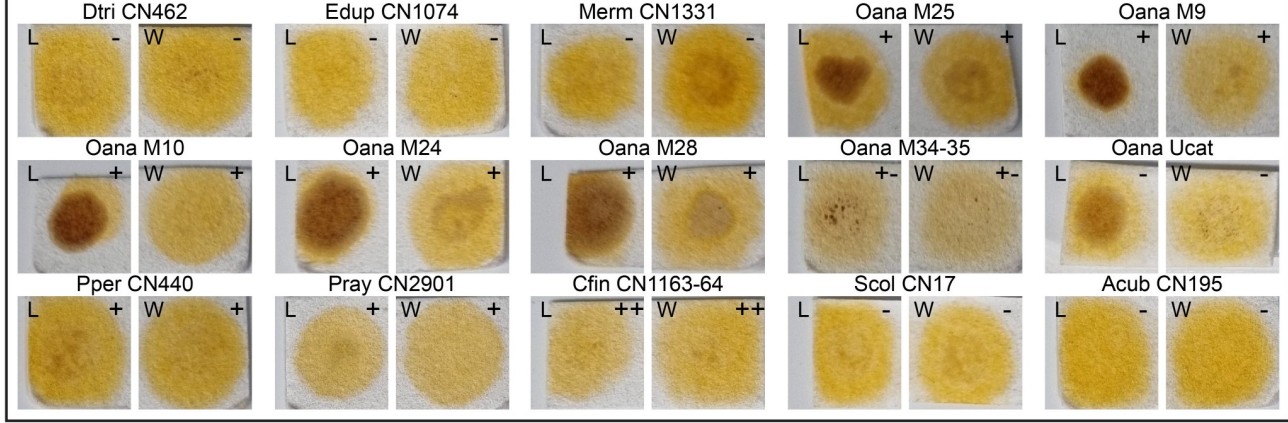

(b)

**As in 3.5x concentrated preservation fluids of specimens with (+) and without (-) nodules**

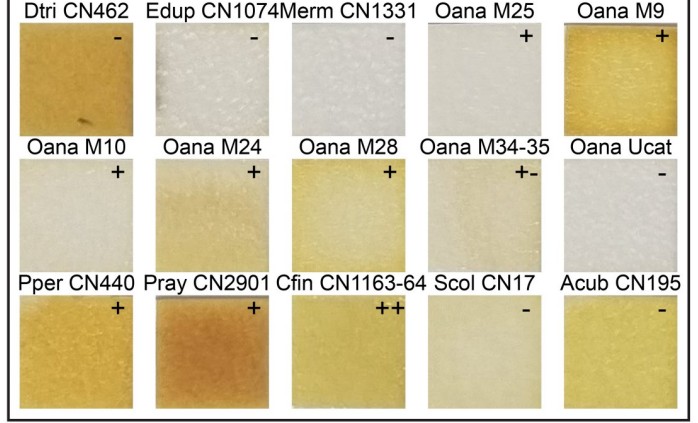

(c)

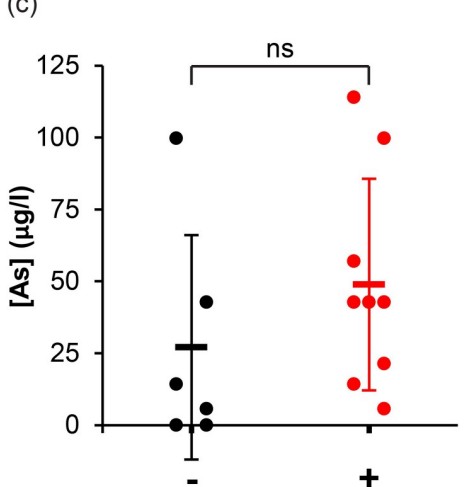

**Fig 6. Chemical analysis of preservation fluids for lead (Pb) and arsenic (As). (a)** Analysis for the presence of Pb in 30x concentrated samples of preservation fluids of mammal specimens (see Table 1 for full names and catalogue#) with (+) or without (-) hyperintense nodules on x-ray images. Some specimens were stored in the same jars as indicated with multiple + or -. Both the lipid (L) and the water (W) phase was analysed but the presence of Pb was not detected in any sample; **(b)** Analysis for the presence of As in 3.5x concentrated samples of preservation fluids (mixed lipid and water phase). In all but two samples As was detected; **(c)** Arsenic concentration in different samples of preservation fluids. There was no significant difference between [As] is samples with and without hyperintense nodules.

modern long-term preservation of biological specimens, and to a lesser degree continued storage in 4% buffered formaldehyde only yielded a small volume of corrosion products within the sample, contact with any unbuffered formaldehyde solution, even followed by transfer to 70% EtOH resulted in much larger volumes of corrosion products (Fig 7D and 7E).

## Radiodensity of lead shot pellets and the effect on measurements of bone mineral content

The Hounsfield scale is a linear quantitative scale for describing radiodensity of CT scanned objects and it is formally defined with distilled water having a CT number of 0 HU and atmospheric air a CT number of -1000 HU and in the 16-bit version it ranges from -32768 to

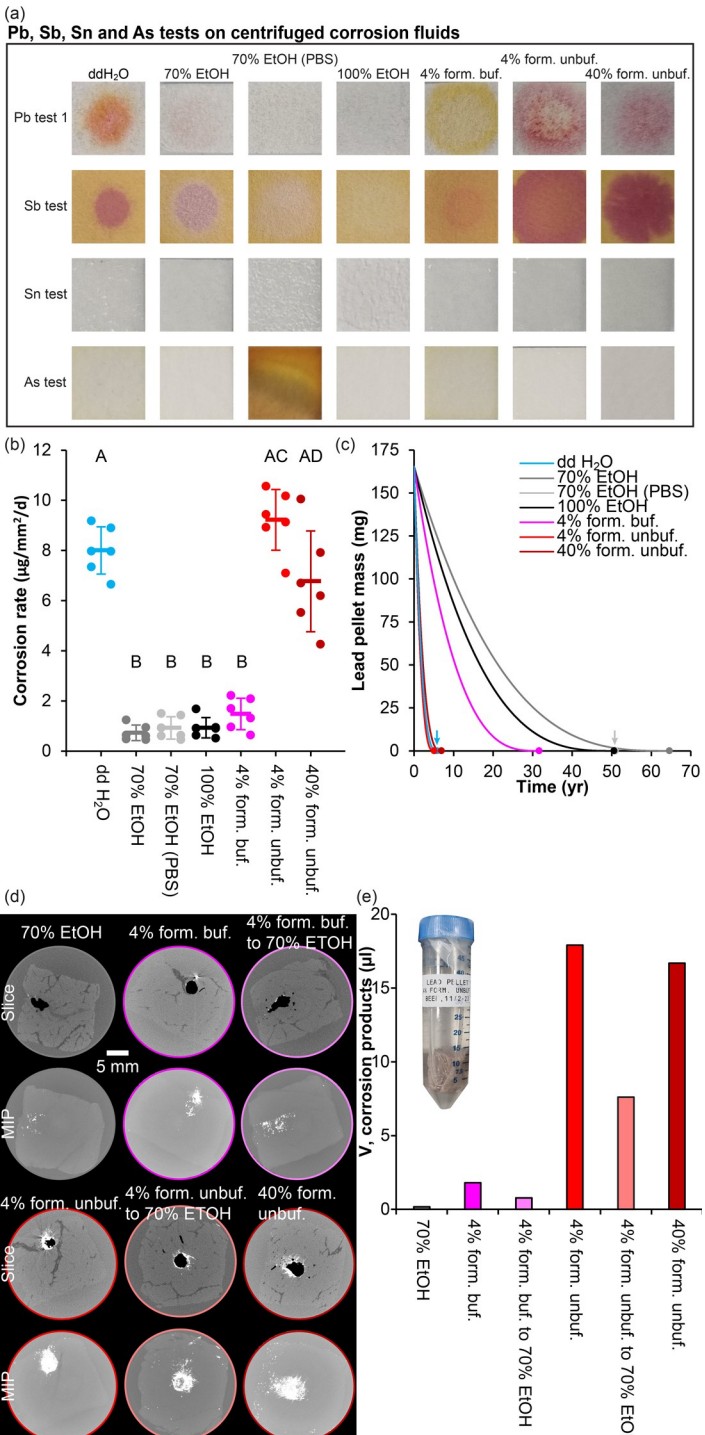

**Fig 7. Corrosion test of lead pellets in different solutions. (a)** Analysis for the presence of lead (Pb), antimony (Sb), tin (Sn) and arsenic (As) after storing lead pellets for 22 days in 1 ml of either double distilled water, 70% ethanol (EtOH) with and without phosphate buffed saline (PBS), 100% EtOH, phosphate buffered 4% formaldehyde and unbuffered 4% and 40% formaldehyde. All but the 70% EtOH (PBS), 100% EtOH and the buffered 4% formaldehyde solutions tested positive for Pb. All solutions except the 100% EtOH tested positive for Sb. Tin was only detected in the unbuffered 40% formaldehyde solution. The water, the buffered 4% formaldehyde and in particular the 70% EtOH solution with PBS contained a detectable amount of As; **(b)** corrosion rate of lead pellets in different solutions. There was significant difference between the corrosion rate in different solutions (one way ANOVA with post hoc Tukey honest significance test). Significantly different groups are marked with different capital letters (A, B, C and D) i.e. a

group labelled with A is significantly different from a groups labelled with B but not from one labelled with AC; **(c)** Lead pellet mass over time at the corrosion rate of different solutions. The point at which all pellet material has been corroded is marked with a circle in the respective line colours (or by an arrow in cases where circles are close); **(d)** Virtual cross sections of beef cubes previously hosting a lead pellet (top rows) or maximum intensity projections (MIP, bottom row). Varying degrees of corrosion products remain within the sample after 8.5 month of storage; **(e)** Volume (V) of corrosion products within beef samples preserved in different preservation fluids.

+32767 HU [27]. It follows that lipid rich tissues are slightly less radiodense than water (-120 to -90 HU), lean tissues are slightly more radiodense than water (0 to +100 HU) and mineral-ised tissues like bone are even more radiodense (+300 to +1900 HU). However, the radiodensity of lead shot pellets surpasses the conventional Hounsfield scale (> 32767 HU) (Fig 8A and 8B). Thus, to estimate the actual Hounsfield value of lead shot pellets, we conducted a profile analysis of free lead pellets, taking advantage of the partial volume effect i.e. the fact that voxels containing both pellet material and surrounding air have averaged Hounsfield values in the measurable range, and thus the actual Hounsfield value of voxels only containing pellet material can be estimated by extrapolating the profile (Fig 8B). The estimated Hounsfield value of lead shot pellets at 120 keVp was 43584 ± 4670 HU. By comparison to the size of the 4320 ± 531 HU foreign objects found in the Dana platypus, this corresponds to the Dana platypus having been shot with lead pellets (disregarding the small density effect of the trace metals Sb, Sn and As within the alloy) with an average diameter of 2.6 ± 0.7 mm equalling a shot size of an English bird shot size #6 (2.59 mm in diameter).

The presence of very radiodense lead shot pellets is of importance when translating acquired CT numbers to measurements of bone mineral density and bone mineral content (BMC) using a calibration phantom with known concentrations of calcium hydroxyapatite. Due to the much higher radiodensity of Pb than Ca (see Fig 3B), measurements of total BMC in a specimen is strongly affected by the presence of metallic Pb and even more so of Pb salts. To demonstrate this, we conducted an experiment on six fresh cadavers of laboratory rats. These were first CT-scanned without any lead shot pellets, and then they were each injected with six lead pellets (Fig 8C). Total BMC was calculated based on a calibration procedure to translate CT numbers to bone mineral density for both the lead shot pellet free state, the state with pellets, and a state in which pellets were digitally removed by enforcing an image threshold of 1900 HU, resulting in most voxels with pellets being removed. Absolute BMC per animal was measured to 5.5 ± 0.4 g in pellet free rats, 8.9 ± 0.4 g in the same rats with pellets, and 5.8 ± 0.4 g when pellets were digitally removed. This was significantly different between all groups (One way ANOVA with repeated measures, $F_{(2,10)}$ = 3917.5, $p < 0.001$ and post hoc Tukey HSD for tests between each group all with $p < 0.001$). Likewise, the body mass adjusted BMC (BMC / body mass fraction) was also significantly different between each group ($F_{(2,10)}$ = 1165.0, $p < 0.001$ and post hoc Tukey HSD for tests between each group all with $p < 0.05$) (Fig 8D). Thus the presence of lead pellets caused a significant overestimation of BMC, and although a digital removal of most voxels within lead shot pellets resulted in a reduction of BMC overestimation (Fig 8D), both absolute BMC and body mass adjusted BMC were still significantly overestimated due to partial volume effects of voxels in the border zone of lead shot pellets that could not be digitally removed by thresholding without also potentially removing actual bone containing voxels (Fig 8C, see magnification of digitally removed pellet in the lower right image with remaining partial volume border zone).

Naturally, the effect of BMC overestimation due to encapsulated lead shot pellets is more dominant in smaller species with a generally low BMC, and it becomes more apparent the higher the number of pellets found in the individual. To demonstrate this, percentage error of the total BMC of animals ranging in BMC from 1–100 g (representing species roughly the size

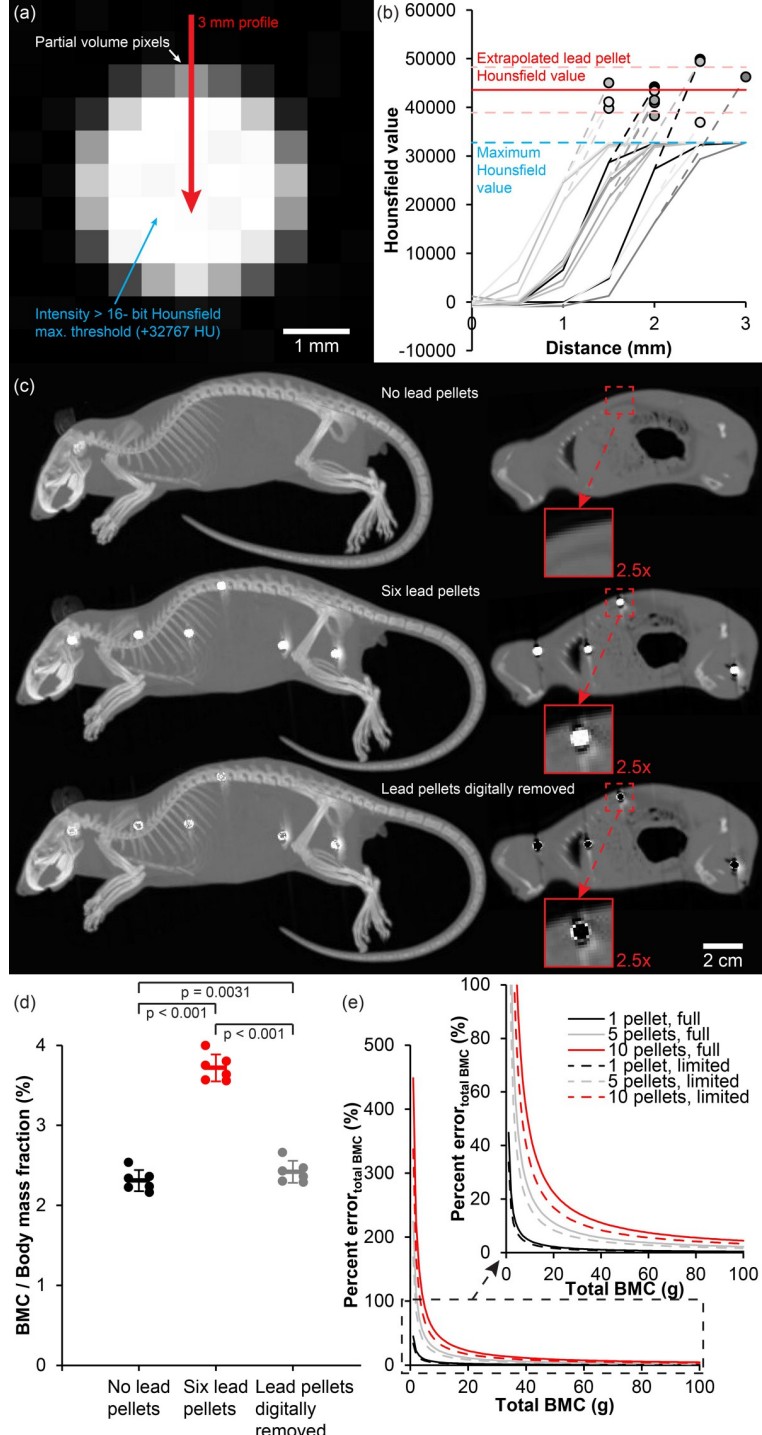

**Fig 8. Determination of lead Hounsfield value and the effect on measurements of bone mineral content in animal specimens. (a)** Virtual section of 3 mm lead shot pellet from x-ray computed tomography (CT). Due to the high radiodensity of the lead alloy, the pellet core exceeds the maximum Hounsfield value on conventional 16-bit CT systems (from -32768 to +32767 HU) and actual radiodensity of lead pellets cannot be measured directly. However, peripheral voxels suffer from partial volume effects from the averaging of signal contribution from lead and surrounding air (-1000 HU); **(b)** By extrapolating the maximum slope of 12 longitudinal profile plots originating in air and terminating at the core of six lead pellets, the partial volume effect can be exploited to indirectly reveal the average radiodensity (43584 ± 4670 HU) of lead pellets allowing for original pellet size calculations of foreign objects in the Dana platypus; **(c)** Laboratory rat before (top) and after (middle) injection of six lead pellets, and after pellets have

been digitally removed by subtracting all voxels with CT values > 1900 HU (bottom). To the left are maximum intensity projections, and to the right are virtual sagittal sections. Red boxes on the right are 2.5x magnifications of a pellet in the chest region; **(d)** Body mass adjusted bone mineral content (BMC) measured using quantitative CT on laboratory rats before and after injection of lead pellets and after digital removal of these. The presence of lead pellets increased body mass adjusted BMC significantly, and even after digital removal of lead pellets, the measurement was significantly affected due to the partial volume voxels with CT-values of a little less than 1900 HU that were still present in the images (see magnification in (c) lower right); **(e)** Percentage error (difference in estimated and actual value) of total BMC as a function of total BMC and the presence of either 1, 5 or 10 lead pellets. Due to the CT-value threshold of 32767 HU on most clinical CT systems, metallic lead alloys of lead shot pellets with CT values of 43584 ± 4670 HU do not express the maximum effect on BMC measurements. However, as lead pellets corrode and radiodensity of lead salts fall below the CT-value threshold, the effect of lead pellets is increased as these corrode. This is demonstrated using solid graph lines for the full effect of imbedded lead pellets and dashed lines for the limited effect when pellets are still in the metallic state.

of laboratory mice to domestic cats) was calculated for 1, 5 and 10 pellets (Fig 8E). Since the radiodensity of the metallic lead alloy found in shotgun pellets surpasses the maximum value of the conventional Hounsfield scale (Fig 8B), the full effect of lead pellets on BMC measurements cannot be observed when pellets are still in the metallic state, since most voxels values within the shot pellet are capped by the maximum Hounsfield value. As lead pellets corrode, however, radiodensity of the corrosion salts falls below the Hounsfield scale maximum, and the full effect of encapsulated lead will act on the BMC measurement. To illustrate this point, we calculated both the full effect (lead pellet Hounsfield value = 43584 HU) and the limited effect (lead pellet Hounsfield value = 32767 HU) of 1, 5 and 10 lead shot pellets imbedded in specimens with BMC in the range of 1–100 g (Fig 8E).

## Discussion

Recently it was estimated that the wet biomass of terrestrial wild mammals (~22 Mt) currently constitute about 2.1% of the biomass of all terrestrial mammals when including humans (~390 Mt) and livestock (~630 Mt) (Greenspoon et al. 2023). Of this, the majority of wild land mammal biomass is found in large-bodied species (>10 kg) such as even-hoofed mammals with a dominance of a few species of deer [28]. In this light, the fluid preserved small mammals in the collections of natural history museums are as important as ever to document past small mammal diversity [6]. However, for anatomical and pathological studies, preserved animals are only as valuable as their collection and preservation record. In this study, we use the Dana platypus to demonstrate a case in which limited information on how a sample was collected could easily have led to erroneous conclusions. At first, the widely distributed large nodules in this specimen bore no resemblance to projectiles (e.g. compare Figs 1B and 3A to Figs 5 and 6 of [29], and Fig 6 in [30]). This indicated a pathological origin. Several pathologies could potentially explain large and dense nodules (Table 2). Although the most obvious aetiology in general would be tuberculosis, since our initial observation of nodules was in a platypus, mucormycosis (also referred to as ulcerative dermatitis, mycotic granulomatous dermatitis and ulcerative mycotic dermatitis was of special interest [31, 32]. This disease caused by the fungus *Mucor amphibiorum* can cause severe granulomatous and ulcerative dermatitis in affected individuals and is a leading cause of morbidity and mortality in wild platypus of Tasmania [30–33]. Although this disease does not usually present with mineralised nodules, the appearance of subdermal and muscular dense nodules in the Dana platypus seemed mostly in agreement with the clinical description of the disease (Table 2). If this had been the case, it would be of significant interest, since the Dana platypus was collected at least 53 years before the first reported observation of mucormycosis in wild platypus in the austral autumn of 1982 [34].

However, after careful analysis we conclude that the nodules in the Dana platypus are foreign objects and not caused by a pathology. When inspected with CT and micro-CT the

nodules appear more radiodense than most mineralised tissues (Fig 1), and the x-ray attenuation spectrum is unlike that of calcified bone (Fig 3). Further, hyperspectral CT revealed a K-edge at 88 keV matching the spectrum of lead (Fig 3G and 3H). A simple flame test revealed that neither an entire nodule nor the core or crust material burned with similar flame colours as the most abundant biometals, Ca, K, Na and Fe (Fig 2A), ruling out some form of naturally occurring precipitation of biometal salts caused by a pathology (e.g., metastatic calcification) or as a result of the preservation process. Instead, both core and crust material contained Pb, Sb, Sn and As (Fig 2B–2D). These metals are uncommon in healthy biological tissues but have for centuries been used in lead alloys for the manufacturing of lead shot pellets: Sb (0.5–6.5% of alloy mass) increases the hardness of lead shot pellets, Sn (~0.1%) increases the malleability of the pellet and reduces fragmentation and As (0.1–0.2%) facilitates the spherical formation of pellets [35–37]. The finding of these four metals strongly suggests the nodules in the Dana platypus to be foreign objects and specifically those of lead shot pellets. This was further supported when inspecting the head of the Dana platypus more closely. A shot pellet trapped in the right cerebral hemisphere had left a trace of fragmented bone and pellet material along the proposed shot trajectory (Fig 4A and 4B), and entry wounds were located at the furless bill when the specimen was left to dry i.e. reducing reflections at the surface (Fig 4C). Finally, similar nodules as in the Dana platypus were observed in six out of the eight fluid preserved platypus in the collection of the Natural History of Denmark collected between 1865 and 1950. In itself, this does not rule out a previous pathology causing mineralised nodules, but it is nonetheless consistent with platypus being harvested by shooting up until 1941–1960 [38]. Also, similar nodules were found in unrelated marsupial and placental mammals in the fluid preserved collection (Fig 5). One of these, the common ringtail possum *Pseudocheirus peregrinus* cat# NHMD-M440, was also collected or donated in Australia during the fourth Dana expedition, and another one, the Solomons flying fox *Pteropus rayneri* cat# NHMD-CN2901, was most certainly collected using a shotgun with a custom-made adapter sleeve (Fig 5B), during the later Noona Dan Expedition. In conclusion, the overwhelming amount of evidence points to the large and irregularly shaped nodules in the Dana platypus and the other inspected mammal specimens being corroded shot pellets. In the Dana platypus, these are with certainty lead shot pellets (Fig 3F–3H). Even though the potentially toxic effects of lead pellets on wildlife were recognised as early as in the late 19[th] century [39, 40], lead alloys were the most popular materials for shot pellet manufacturing up until a few decades ago [41]. Accordingly, the pellets found in the remainder of the studied specimens are most likely also lead shot pellets.

Although collection practises using firearms are well known, and often associated with high collection outputs e.g. in the collection of birds, the use of shotguns can increase outputs by up to 50% compared with the use of traps and nets (Hosner, P.A., Natural History Museum of Denmark, personal communication), the potential effects of preserving specimens containing shot pellets remain underreported. Apart from a brief mention in an ornithological protocol paper of the preference of collecting bird specimens for fluid collections with other methods than shooting since shot pellets can interfere with modern imaging techniques [42], and the interesting discovery of very small lead pellets in the skull of the best preserved dodo specimen in existence, the Oxford dodo [43], we have not been able to locate any other relevant literature on this subject apart from a mention of the corrosion of metal tags in fluid preserved collections [3, 44]. Veterinary forensic pathology, while an increasing speciality, deals usually with recent projectile impacts with different characteristic appearance to the corroded tissue embedded features described here [29], or lead shot or bullets as a toxicology hazard when ingested, fragmented and absorbed in the gastrointestinal tract of wildlife (e.g. [22, 45]. Apart from the risk of misinterpreting corroded shots as pathologies there are two potential concerns

when handling affected specimens and including these in comparative studies: The potential risk of unknowingly handling toxic metals and the effect on quantitative measurements.

Both Pb and As are toxic metals, and both are recognised as elements of concern in museum collections introduced via collection items such as lead objects and pigments or previously used preservation substances e.g. arsenic soaps [46–49]. The lead shot pellets found in the Dana platypus and additional specimens were clearly no longer in the metallic state (Fig 1C and 1D), but consisted of corrosion products much larger than the original shot pellets (Table 1 and Fig 5). The calculated average size of original shot pellets of 2.6 ± 0.7 mm (English bird shot size #6) in the Dana platypus seems to match a reasonable shot size for dispatching an animal of that size, in particular since the large number of incorporated pellets indicates a fairly short shot distance with a tight grouping. Thus, only a limited amount of pellet material would be expected to have dissolved in the preservation fluid. In regards of Pb, this was also the case, since no trace of Pb was detected in upconcentrated preservation fluids of any of the included samples, demonstrating that the concentrations of Pb in the original samples were < 1.2 µM (Fig 6A). Even at high temperatures, water solubility of the lead salt which made up the nodules was low (Fig 2E and 2F) indicating that the corrosion product did not consist of lead(II) acetate or lead(II) formate, which have been suggested as common corrosion products of lead in museum collection [50], nor a precipitation of lead(II) chloride after reaction with widely present chloride ions in the tissue. Most likely the corroded lead pellets consisted of another highly insoluble corrosion product such as e.g. lead(II) oxide, lead(IV) oxide, lead(II) carbonate, lead (II) phosphate or a mixture. Small particles were encapsulated in the fur of the entire specimen which could be observed with CT (S1 File) and micro-CT imaging (Fig 3F, arrow). At first there was a concern that these particles could be precipitated Pb salts, however hyperspectral CT with K-edge subtraction revealed no traces of Pb in these particles (Fig 3F, arrow), which are more likely to consist of sand grains. In the case of As, traces of this metal were found in preservation fluids of both specimens with and without shot pellets with no significant differences (Fig 6C). This may be a result of As leakage from other sources e.g. previously used arsenic soaps, or due to repeated use and mixing of preservation fluids between samples over time. Although As exposure should be minimised using personal protection equipment such as nitrile gloves [45], even the highest concentration of 114.0 µg/l found in the preservation fluid of *Pteropus rayneri* cat# NHMD- M05-CN2901 (Fig 6B and 6C), is well below the 500 µg/l threshold considered safe for bathing/showering in household water in many places [51, 52]. In summary, our analysis showed that handling biological specimens containing corroded lead pellets does not directly expose collection staff to large concentrations of Pb and As, but care is warranted if performing dissection which could directly expose encapsulated lead nodules.

Our results highlight that the preservation history of museum specimens may be of importance for the corrosion of embedded shot pellets. By measuring corrosion rate of lead shot pellets, we found that solutions with a close to neutral or slightly alkaline pH (70% EtOH, pH = 6.03; buffered 4% formaldehyde, pH = 6.84; 70% EtOH (PBS), pH = 8.61) or a water free solution (100% EtOH) displayed a significantly lower rate of corrosion on lead pellets than more acidic and unbuffered solutions (double distilled water, pH = 5.01; unbuffered 4% formaldehyde, pH = 3.97; unbuffered 40% formaldehyde, pH = 3.18) (Fig 7B). This translates to a much faster corrosion of internalised lead pellets if specimens are stored in e.g. unbuffered 4% formaldehyde than 70% EtOH (Fig 7C–7E). However, when approaching and surpassing centuries of storage, lead pellets may eventually corrode fully (Fig 5A). It is of interest to note, however, that no clear pattern of age and the degree of corrosion is evident in the studied specimens e.g. compare *Ornithorhynchus anatinus* NHMD-M01-10 collected in 1865 and *Ornithorhynchus anatinus* NHMD-M01-25 collected in 1915, both displaying large and highly

corroded pellets, to *Ornithorhynchus anatinus* NHMD-M01-9 collected in 1865 with much smaller pellets (Fig 5A). Likewise, the introduction of formaldehyde fixation following the discovery in 1893 (the first reported use at the Natural History Museum of Denmark was in 1902 [53]), also does not seem to influence the corrosion state of pellets (Fig 5A). In opposition to Pb, the corrosion test revealed that storing lead pellets in slightly alkaline 70% EtOH (PBS) containing sodium phosphate salts resulted in a much higher concentration of As in the preservation fluid compared to other solutions (Fig 7A). It has been suggested to replace the pure water content of widely used 70% EtOH in museum collections with more physiological relevant salt solutions for long-term storage of fluid preserved specimens [21], however in cases of specimens containing lead shot pellets our results demonstrate that caution is warranted, since this could lead to increased solubility of As.

The other major concern of lead shot pellets in fluid preserved specimens comes into play when using these specimens for quantitative imaging analyses of tissue components in comparative studies. In our experience, mineralised bony material is well-preserved in well-managed fluid collections although some demineralisation can take place in unbuffered preservation fluids over time. Thus, fluid preserved specimens are useful in studies of comparative osteology. Important parameters when comparing skeletons are bone mineral density and BMC. These parameters are directly linked to bone strength and for semi-aquatic and fully aquatic species mineral content plays an important role in buoyancy regulation [11]. Traditionally, BMC is measured by ashing i.e. heating bone samples to >500°C to burn off all organic matter, but quantitative CT allows for a much faster and non-destructive method of estimating both BMC in addition to density and volume of bones of interest [11]. However, this method is sensitive to the presence of foreign objects with a much higher radiodensity than calcified bones. In this study, we estimated radiodensity of lead shot pellets to 43584 ± 4670 HU. This is ~23 times denser than the densest bones in the human organism, and most likely even more so in smaller mammals considering allometric scaling of skeletal strength in terrestrial mammals [54]. This means that, if overlooked, a single or just a few lead shot pellets can have significant effect on the estimation of BMC in small mammals (Fig 8C–8E). At increasing body sizes and bone mineral contents, this effect is decreased (Fig 8E), but lead pellets are also easier to overlook in large specimens. If pellets are recognised and they remain in the very radiodense metallic state, their effect on BMC can be alleviated to some degree by applying a low pass image filter e.g. removing all pixels with Hounsfield values > 1900 HU. But due to the partial volume effect at the interface between pellets and surrounding tissue, the BMC estimate will still be affected (Fig 8C and 8D). If pellets are corroded, this task becomes even more difficult since the radiodensity of corrosion products of Pb is considerably closer to the radiodensity of calcified bone (Fig 1G), making image filtering procedures much more challenging or even unsuited. In these cases, affected specimens may be excluded for analysis.

In summary, a coincidental observation of dense nodules in an old fluid preserved specimen of platypus of the mammal collection at the Natural History Museum of Denmark inspired us to thoroughly examine the origin of these structures. This work led to an increased understanding of the behaviour of corroding lead shot pellets during long-term preservation of museum specimens, and potential caveats when using affected specimens to draw patho-anatomical veterinary and osteology information. Most of the applied analysis methods are simple and inexpensive, e.g. flame test and metal test papers and kits, and are applicable to test samples for the presence of lead shot pellets at museums and research institutions throughout the world. These simple methods can be complemented with more sophisticated methods such as quantitative CT and hyperspectral CT based on availability. Of importance, we found that leakage of toxic Pb and As from corroding lead shot pellets may not constitute a direct

hazard when handling intact specimens and their preservation fluids. However, caution is warranted especially if dissecting previously shot specimens, thus exposing toxic metal salts. In such cases, and before undertaking large comparative analysis sensitive to the presence of lead pellets, high throughput screening for residual foreign objects using e.g. digital x-ray imaging should be considered. Further studies should investigate the fate of the steel, bismuth, and tungsten pellets that have in some places replaced lead in shotgun pellets.

## Supporting information

**S1 File. Collection of photos, images, slice videos and virtual renders produced from CT and MRI.** This material was originally presented to highly trained research and zoo veterinarians (authors CJAW, DS, AKOA and MFB) to form an opinion on what disease the hyperintense nodules in the Dana platypus could originate from.
(ZIP)

## Acknowledgments

We wish to thank the Department of Forensic Medicine, Aarhus University, specifically Professor Lene Warner Thorup Boel for kindly providing access to the x-ray computed tomography system. Likewise, we wish to thank the MR-research Center, Aarhus University, specifically Professor Christoffer Laustsen for providing affordable access to magnetic resonance imaging. We also wish to thank the Clinic for Osteoporosis, Hormone and Bone Diseases, Aarhus University Hospital, specifically Lars Rejnmark and Tove Stenum for providing access to the extremity CT system used to image nodules. Finally, we wish to thank Daniel Kjær Andersen for help with the rat cadaver experiment.

## Author Contributions

**Conceptualization:** Henrik Lauridsen, Daniel Klingberg Johansson, Christina Carøe Ejlskov Pedersen, Kasper Hansen, Peter Rask Møller.

**Data curation:** Henrik Lauridsen.

**Formal analysis:** Henrik Lauridsen, Catherine Jane Alexandra Williams, Ditte-Mari Sandgreen, Aage Kristian Olsen Alstrup, Mads Frost Bertelsen, Peter Rask Møller.

**Funding acquisition:** Henrik Lauridsen, Peter Rask Møller.

**Investigation:** Henrik Lauridsen, Daniel Klingberg Johansson, Christina Carøe Ejlskov Pedersen, Kasper Hansen, Michiel Krols, Kristian Murphy Gregersen, Julie Nogel Jæger, Catherine Jane Alexandra Williams, Ditte-Mari Sandgreen, Aage Kristian Olsen Alstrup, Mads Frost Bertelsen, Peter Rask Møller.

**Methodology:** Henrik Lauridsen, Daniel Klingberg Johansson, Michiel Krols, Kristian Murphy Gregersen, Julie Nogel Jæger.

**Project administration:** Henrik Lauridsen.

**Resources:** Henrik Lauridsen, Peter Rask Møller.

**Software:** Henrik Lauridsen.

**Supervision:** Peter Rask Møller.

**Validation:** Henrik Lauridsen.

**Visualization:** Henrik Lauridsen, Peter Rask Møller.

**Writing – original draft:** Henrik Lauridsen.

**Writing – review & editing:** Daniel Klingberg Johansson, Christina Carøe Ejlskov Pedersen, Kasper Hansen, Michiel Krols, Kristian Murphy Gregersen, Julie Nogel Jæger, Catherine Jane Alexandra Williams, Ditte-Mari Sandgreen, Aage Kristian Olsen Alstrup, Mads Frost Bertelsen, Peter Rask Møller.

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
