## [Decision Letter · Decision Letter 0]

12 Aug 2024

PONE-D-24-21732The curious case of the Dana platypus and what it can teach us about how lead shotgun pellets behave in fluid preserved museum specimens and may limit their scientific valuePLOS ONE

Dear Dr. Lauridsen,

Thank you for submitting your manuscript to PLOS ONE. After careful consideration, we feel that it has merit but does not fully meet PLOS ONE’s publication criteria as it currently stands. Therefore, we invite you to submit a revised version of the manuscript that addresses the points raised during the review process.

We look forward to receiving your revised manuscript.

Kind regards,

Gianniantonio Domina, Ph.D.

Academic Editor

PLOS ONE

Journal Requirements:

2. Thank you for stating the following financial disclosure: "HL is supported by the Carlsberg Foundation (grant# CF21-0605) and PRM is supported by the Carlsberg Foundation (grant# CF21-0435)."

3. Thank you for stating the following in the Competing Interests section: "I have read the journal's policy and the author Michiel Krols of this manuscript has the following competing interests: Employed at TESCAN XRE, a manufacture of one of the micro-CT systems used in the study (UniTOM XL Spectral)."

Additional Editor Comments:

Dear Dr. Lauridsen,

the reviewers suggested that your manuscript deserves to be accepted for publication after minor, formal review.

The argument deserves to be published in Plos ONE, the methdology used is suitable with your aims.The form is good.

Please, follow the suggestions of reviewer one to prepare your improved MS.

Best Wishes

Gianniantonio Domina

Reviewers' comments:

Reviewer's Responses to Questions

**Comments to the Author**

1. Is the manuscript technically sound, and do the data support the conclusions?

Reviewer #1: Yes

Reviewer #2: Yes

2. Has the statistical analysis been performed appropriately and rigorously? 

Reviewer #1: Yes

Reviewer #2: Yes

3. Have the authors made all data underlying the findings in their manuscript fully available?

Reviewer #1: Yes

Reviewer #2: Yes

4. Is the manuscript presented in an intelligible fashion and written in standard English?

Reviewer #1: Yes

Reviewer #2: Yes

5. Review Comments to the Author

Reviewer #1: To the authors:

1/ The manuscript is technically sound, and the offered data support the conclusions

2/ The statistical analysis have been performed appropriately and rigorously

3/ The authors provide all data in the text or in the supplementary data to suport the findings in their manuscript

4/ The English is ok

Comments:

a/ Congratulations on the focus on spirit Natural History Collections.

b/ Just some minor corrections, including in the references secction

Reviewer #2: Very useful, interesting, and well written.

Since fornaldehyde is a gas at room temperature the expression 4% v/v formaldehyde in misleading. It should be replaced by 3.7% w/v formaldehyde or approximately 4% formaldehyde or 10% v/v formalin (formalin is 37% w/v formadehyde). See: Finkelde, I. and R.R. Waller. 2020. Comparing Methods of Determining Formalin Concentration in Fluid Preservatives. Collection Forum 34 (1): 32–52. https://doi.org/10.14351/0831-4985-34.1.32

I expect the most likely lead corrosion product would be lead phosphate. It is unfortunate that X-ray diffraction analysis of the corrosion products could not have been done. Unfortunate also that more sensitive methods of preservative fluid analysis, such as atomic absorption analysis were not employed.

These are minor points and overall the manuscript is very good and worthy of publication.

6. PLOS authors have the option to publish the peer review history of their article (what does this mean?). If published, this will include your full peer review and any attached files.

Reviewer #1: No

Reviewer #2: **Yes: **Robert Waller

---

## [Editor Report · Decision Letter 1]

20 Aug 2024

The curious case of the Dana platypus and what it can teach us about how lead shotgun pellets behave in fluid preserved museum specimens and may limit their scientific value

PONE-D-24-21732R1

Dear Dr. Lauridsen,

We’re pleased to inform you that your manuscript has been judged scientifically suitable for publication and will be formally accepted for publication once it meets all outstanding technical requirements.

Kind regards,

Gianniantonio Domina, Ph.D.

Academic Editor

PLOS ONE

Additional Editor Comments (optional):

The authors have implemented some few minor corrections. The Ms is ready to be paged.
---

## [Editor Report · Acceptance letter]

22 Aug 2024

PONE-D-24-21732R1 

PLOS ONE

Dear Dr. Lauridsen, 

I'm pleased to inform you that your manuscript has been deemed suitable for publication in PLOS ONE. Congratulations! Your manuscript is now being handed over to our production team.

Kind regards, 

on behalf of

Prof. Gianniantonio Domina 

Academic Editor

PLOS ONE